# Prevalence and incidence of neuromuscular conditions in the UK between 2000 and 2019: A retrospective study using primary care data

Iain M. Carey[ID][1]*, Emma Banchoff[1] , Niranjanan Nirmalananthan[2] , Tess Harris[1] , Stephen DeWilde[1] , Umar A. R. Chaudhry[1] , Derek G. Cook[1]

1 Population Health Research Institute, St George's, University of London, London, United Kingdom,
2 Department of Neurology, St George's Hospital NHS Trust, London, United Kingdom

These authors contributed equally to this work.
* sgjd450@sgul.ac.uk

## Abstract

### Background

In the UK, large-scale electronic primary care datasets can provide up-to-date, accurate epidemiological information on rarer diseases, where specialist diagnoses from hospital discharges and clinic letters are generally well recorded and electronically searchable. Current estimates of the number of people living with neuromuscular disease (NMD) have largely been based on secondary care data sources and lacked direct denominators.

### Objective

To estimate trends in the recording of neuromuscular disease in UK primary care between 2000–2019.

### Methods

The Clinical Practice Research Datalink (CPRD) database was searched electronically to estimate incidence and prevalence rates (per 100,000) for a range of NMDs in each year. To compare trends over time, rates were age standardised to the most recent CPRD population (2019).

### Results

Approximately 13 million patients were actively registered in each year. By 2019, 28,230 active patients had ever received a NMD diagnosis (223.6), which was higher among males (239.0) than females (208.3). The most common classifications were Guillain-Barre syndrome (40.1), myasthenia gravis (33.7), muscular dystrophy (29.5), Charcot-Marie-Tooth (29.5) and inflammatory myopathies (25.0). Since 2000, overall prevalence grew by 63%, with the largest increases seen at older ages (≥65-years). However, overall incidence remained constant, though myasthenia gravis incidence has risen steadily since 2008, while new cases of muscular dystrophy fell over the same period.

**Data Availability Statement:** The study is based on electronic health data from the Clinical Practice Research Datalink (CPRD) obtained under license

from the UK Medicines and Healthcare Products Regulatory Agency (MHRA). CPRD data governance and the license to use CPRD data does not allow distribution of patient data directly to other parties. Researchers must apply directly to CPRD for data access (https://www.cprd.com).

**Funding:** The study was funded from a research grant from Muscular Dystrophy UK (Grant Ref 19GROE-PG24-0349-2) awarded to Iain Carey as principal investigator. The funders had no role in the study design, data collection and analysis, decision to publish or preparation of the manuscript.

**Competing interests:** The authors have declared that no competing interests exist.

## Conclusions

Lifetime recording of many NMDs on primary care records exceed current estimates of people living with these conditions; these are important data for health service and care planning. Temporal trends suggest this number is steadily increasing, and while this may partially be due to better recording, it cannot be simply explained by new cases, as incidence remained constant. The increase in prevalence among older ages suggests increases in life expectancy among those living with NMDs may have occurred.

## Introduction

Neuromuscular disease (NMD) comprises a group of individually rare conditions which affect muscle functioning [1]. While many are hereditary and life-limiting, such as Duchenne muscular dystrophy [2], others are autoimmune conditions often acquired in adulthood, such as Guillain-Barré syndrome, and can be successfully treated [3]. Whilst each condition can be classified as rare; in Europe this is defined as affecting no more than 1 in 2,000 [4], as a combined entity they may represent a more significant group, with a similar population prevalence to other neurological diseases such as Parkinson's and multiple sclerosis [5].

Patients with a NMD require specialist and often long-term multi–disciplinary care to monitor and manage their health. In the UK, access to such specialist services, community therapies and social care vary by region, resulting in potential care inequities [6]. While there has been a large increase in the number of people recorded as dying with NMDs in England [7], accurate estimates for how many are living with a NMD in the UK have been lacking and national data have long been sought [8]. A better understanding of the epidemiology of NMD would help estimate both current and potential future healthcare burdens, especially if the number of people living with NMD is rising [9].

Estimates of the prevalence and incidence of NMD have generally used secondary care datasets or disease registries, which lack direct denominator data [8]. In the UK, large-scale electronic primary care datasets provide up-to-date, accurate epidemiological information on a wide spectrum of diseases and conditions [10]. For rarer diseases such as NMD, which are not usually diagnosed in primary care, specialist diagnoses from hospital discharges and clinic letters are reliably transferred on to primary care computerised systems and are electronically searchable. Our aim then, was to use a large UK primary care database to estimate recent trends in the prevalence and incidence of NMD.

## Materials and methods

### Data source

The Clinical Practice Research Datalink (CPRD) is a primary care database in the UK jointly sponsored by the Medicines and Healthcare products Regulatory Agency and the National Institute for Health Research [11]. Research using CPRD data has helped inform clinical guidance and best practice and resulted in over 2,000 peer-reviewed publications. Previously, a limitation was that it was based only on practices using the Vision software system (CPRD GOLD), which has limited market share in some UK regions [12]. However, CPRD now includes practices using EMIS software (CPRD Aurum [13]), resulting in a more representative and larger dataset (>1,800 combined practices, 65 million patient lives, of which 16 million are currently active).

We combined GOLD and Aurum datasets into a single dataset to estimate prevalence and incidence rates over time. While practices include full historical records for their patients, we only included a practice's data into our analysis from the point they satisfy proxy data quality measures based on annual deaths and denominator counts. Where the same practice appeared in both datasets and overlapped for some periods, we selected the dataset with the most recent extraction to ensure no duplication. Using data extracted in December 2019, this resulted in the inclusion of 1,745 unique practices (n = 1,036 Aurum, n = 724 GOLD) which contributed at least one year's data during the study period; on average 1,460 practices being active in any single year (Table 1). In these practices there were approximately 33 million patients (Aurum = 22 million, GOLD = 11 million) registered between 1/1/2000 and 1/1/2019. A total of 1,418 practices were still actively providing data as of 1/1/2019, with the majority having data to the end of October 2019.

We included patients in our denominator if they were (i) classified as acceptable by CPRD internal algorithm, and (ii) had accrued 90 days registration time (98% of eligible patients). The latter was done to minimise the impact of any historical diagnoses erroneously appearing on the record soon after registration. The only exception was for patients born and registered in the same year; since diagnoses at birth would be correctly counted as incident—these patients were included from registration date.

**Table 1. Summary of annual denominators in harmonised CPRD dataset.**

| Year | Number of eligible practices on each Jan 1st | Number of eligible patients on each Jan 1st * | Total person-years† |
|------|------|------|------|
| 2000 | 1,016 | 7,946,282 | 8,195,682.9 |
| 2001 | 1,126 | 8,750,769 | 9,005,712.2 |
| 2002 | 1,257 | 9,646,707 | 9,910,890.6 |
| 2003 | 1,338 | 10,323,460 | 10,671,952.9 |
| 2004 | 1,443 | 11,231,096 | 11,430,047.9 |
| 2005 | 1,509 | 11,764,303 | 11,872,850.6 |
| 2006 | 1,548 | 12,093,891 | 12,142,267.5 |
| 2007 | 1,559 | 12,277,606 | 12,354,570.1 |
| 2008 | 1,579 | 12,534,330 | 12,596,132.3 |
| 2009 | 1,589 | 12,727,860 | 12,780,457.9 |
| 2010 | 1,591 | 12,873,421 | 12,868,733.2 |
| 2011 | 1,591 | 12,966,904 | 12,958,730.7 |
| 2012 | 1,584 | 12,979,496 | 13,034,808.3 |
| 2013 | 1,576 | 13,085,197 | 12,882,616.5 |
| 2014 | 1,553 | 12,768,939 | 12,723,483.5 |
| 2015 | 1,524 | 12,729,339 | 12,683,289.4 |
| 2016 | 1,485 | 12,586,031 | 12,553,122.1 |
| 2017 | 1,460 | 12,550,360 | 12,598,056.9 |
| 2018 | 1,447 | 12,661,474 | 12,660,989.2 |
| 2019 | 1,418 | 12,625,287 | 10,424,268.1 |

* —To be eligible patient had to be actively registered on Jan 1st for at least 90 days (unless born and registered in the prior year).

† —Person-years calculated over the entire year. It can be greater than number of eligible patients if practices have entered the dataset during the year, or less if practices leave the dataset during the year. No data was available beyond until mid-November in 2019.

## Incidence and prevalence

For each year in our study (2000–2019), we calculated prevalence and incidence rates (per 100,000 persons or person-years respectively).

**Prevalence.** For the denominator, we calculated the total number of patients aged 0–99 actively registered on January 1st in each year (with 90 accrued days of registration). As only year of birth was available across the whole dataset, age was estimated on January 1st based on their age in the prior year. To be a prevalent case from this group, a prior diagnosis had to be present anywhere on the patient record at that time.

**Incidence.** Person-time registration days was estimated for each year, as the proportion of days in the year they were actively registered (ignoring the initial 90 days of their registration). Incident cases were identified where the first recorded diagnosis occurred in the person-time period for the year, and an incidence rate was estimated excluding patients from the denominator who had an existing diagnosis at the beginning of their person-time period for that year.

## Definition of neuromuscular disease

Diagnoses are recorded on CPRD using a hierarchical clinical classification system called Read codes [14], from clinical sources such as hospital discharge summaries or communication from specialists. We first categorised NMD into 4 broad categories: (i) Motor Neuron Disorders, (ii) Muscle Disease (excluding rhabdomyolysis), (iii) Neuropathies (excluding acquired non-immune) and (iv) Neuromuscular Junction Disorders. Within each category we sub-divided further where Read codes existed and were used such that we could consistently classify the condition within the data (S1 Table). For example, while Read codes exist for individual muscular dystrophies, these were not used consistently over time, so we summarised this group as "Muscular Dystrophy". However, we carried out a supplementary analysis summarising the most common specific muscular dystrophies for the most recent estimates.

In total, we created 22 different groups to classify patients based on the Read codes in their record (S2 Table). Patients could belong to multiple categories, except for "unspecified" groups where they were only classified here if no other specific codes had been recorded. Motor neuron disorders were divided into spinal muscular atrophy, motor neurone disease and post-polio syndrome. Muscle disease was divided into acquired myopathies (endocrine, infectious, inflammatory, toxic or drug-induced), hereditary myopathies (congenital myopathies, metabolic myopathies, muscular dystrophy), mitochondrial disease and muscle channelopathies (non-dystrophic myotonic disorders, periodic paralysis). Hereditary neuropathies were classed as Charcot-Marie Tooth disease or other, while inflammatory & autoimmune neuropathies were classed as Guillain-Barré syndrome or other. Neuromuscular junction disorders were grouped as Eaton-Lambert syndrome, myasthenia gravis or other. Finally, we included a non-specific category ("Muscular or neuromuscular disease unspecified") as some Read codes would not allow clear classification into any other category.

While we did not carry out any within patient validation of diagnoses, we were vigilant to any erroneous recording in the dataset. In the process of extracting and assembling the codes for NMD we made two further pragmatic exclusions. Firstly, we did not count any specific Read codes for Duchenne or Becker muscular dystrophy among females. While Read codes exist to indicate the patient is a carrier of the genetic mutation, it appeared the diagnostic code for the condition was sometimes being used in error. We also excluded Read codes for medium-chain acyl-CoA dehydrogenase deficiency (MCADD) in 3 practices which had unusually high counts for unknown reasons.

## Statistical analyses

To summarise the most recent figures, we estimated prevalence rates per 100,000 persons (with 95% confidence intervals) overall, and for men and women separately, based on a date of 1/1/2019. For incidence, we estimated rates per 100,000 person-years using 5 years' worth of data during 2015–9. We present both crude estimates, and age-sex standardised rates based on Office for National Statistics (ONS) mid-year population estimates in 2019 for the whole of the UK [15]. We also estimated and compared rates for overall NMD by country (England, Scotland, Wales, Northern Ireland) and by deprivation (England only) by indirectly age-standardising to the overall CPRD population using 5-year age groups. For deprivation, the reference population was restricted to all patients with a linked deprivation score.

To compare trends over time, rates were now directly age standardised using weights based on the CPRD population as of 1/1/2019 using 5-year age groups. When we estimated trends within broader age categories (0–14, 15–44, 45–64, 65+), we continued to age standardise to the relevant 2019 population. When comparing trends within specific conditions, we focused on the 6 conditions with the highest incidence/prevalence rates (motor neurone disease (MND), idiopathic inflammatory myopathies (IIM), muscular dystrophies (MD), Charcot-Marie Tooth disease (CMT), Guillain-Barré syndrome (GBS) and myasthenia gravis (MG)). Patients could belong to multiple categories for summary estimates except for an analysis where we looked at the relative contribution of the conditions to the overall prevalence. Here patients were assigned to only one category based on the most recent recorded Read code. As a high percentage (70%) of patients diagnosed with GBS are reported to experience full recovery [16], we report both a lifetime and a 5-year period prevalence based on any recurring codes in their record. Similarly, we present two estimates for prevalence of all NMD based on whether lifetime or 5-year period prevalence of GBS is included.

## Ethics approval

This study is based in part on data from the Clinical Practice Research Datalink obtained under licence from the UK Medicines and Healthcare products Regulatory Agency. The protocol (no. 19_211) was approved by the Independent Scientific Advisory Committee evaluation of joint protocols of research involving CPRD data in October 2019. The approval allows analysis of anonymous electronic patient data without the need for written or oral consent.

## Results

### Available denominators

Table 1 summarises the number of eligible patients and patient-years registration time in each year between 2000 and 2019. Earlier years had fewer practices available due to concerns around data quality, but from 2006 at least 12 million patients were actively registered on subsequent January 1st dates. By 2019, a total of 12,625,287 patients were actively registered, with geographical spread by country as follows: 83% England, 9% Scotland, 7% Wales, 2% Northern Ireland. The number of estimated patient-years declines in 2019 as the data extract did not extend beyond October 2019.

### Prevalence and incidence summary

By the beginning of 2019 (Table 1), 28,230 active patients in CPRD had ever received a NMD diagnosis (prevalence rate = 223.6 per 100,000 persons, 95%CI 221.0–226.2), which was higher among males (239.0) than females (208.3). Standardising to mid-year population estimates for the UK produced a lifetime prevalence estimate of 220.3 (95%CI 217.7–222.9). Among our

classifications, GBS (40.1) had the highest lifetime prevalence, followed by MG (33.7), CMT (29.5), MD (29.5), and IIM (25.0). Only about 1-in-4 patients with a history of GBS had a code recorded in the last 5 years. Excluding patients who had only had older GBS codes lowered the overall (crude) prevalence of NMD to 194.6 per 100,000 persons (95%CI 192.1–197.0). Among specific muscular dystrophies (where recorded), the most common diagnosis was for (Type 1) myotonic dystrophy (S3 Table). Of the 28,230 individuals with any NMD recorded by 2019, 778 (2.8%) received multiple classifications in Table 2, with almost half of these patients (342, 44%) having a Read code indicating muscular dystrophy.

Fig 1 (with accompanying data in S4 Table) sub-divides the 2019 prevalence for any NMD by 5-year age group and type of NMD (patients only appear once in the chart and the sum of the stacked bars represents the total prevalence in each age group). The figure demonstrates the steady rise of NMD with age, peaking at 80–84 years (530.6 lifetime, 457.3 including GBS recorded in last 5 years only). MG is the biggest factor behind this trend, being rare in under 40's but contributing greatly at older ages (143.9 at age 80–84). The "other" category changed with age. At ages 0–4, spinal muscular atrophy and metabolic myopathies (e.g. MCADD) were the most common conditions within the "other" category, while at older ages (>80) the most common were post-polio syndrome, other inflammatory and autoimmune neuropathies (e.g., neuralgic amyotrophy) and toxic or drug-induced myopathy.

For incidence during 2015–9 (Table 3), a total of 8,563 patients received a first code for any NMD, representing an estimated rate of 14.1 per 100,000 person-years (95%CI 13.8–14.4), which was higher in males (15.8 vs 12.4). Standardising to mid-year population estimates for the UK produced an incidence estimate of 14.2 (95%CI 13.9–14.5). Among individual conditions, the highest estimated incidence was for MND (3.4), followed by MG (2.5) and GBS (1.7). For the most frequent conditions, males generally had higher incidence rates with the exception of IIM.

## Variations by region and deprivation

We explored variation in recent prevalence and incidence rates by region (S5 and S6 Tables) and IMD (S7 and S8 Tables). There was no consistent pattern by region, such that regions with the highest or lowest overall prevalence or incidence—reported some conditions higher with others being lower than the rest of the UK. Although numbers of cases were small in some of these regions, there were no obvious outliers in terms of incidence or prevalence rates. For deprivation in England (using IMD), the most consistent pattern was with Guillain-Barré syndrome, where both recent incidence (23% higher than expected) and recorded lifetime prevalence (8% higher) was greatest in the least deprived group.

## Time trends in overall prevalence and incidence

The annual trends in prevalence and incidence between 2000 and 2019 for all NMD for females (Fig 2A) and males (Fig 2B) separately, age-standardised to the 2019 population were estimated (accompanying data in S9 and S10 Tables). For both sexes, there was a clear increasing trend in prevalence over time (66% increase for males, 61% for females for lifetime prevalence). By contrast, overall incidence was largely unchanged over the 20-year period, though in males it was slightly raised during 2016–7. Over the study period the median age among prevalent cases had risen by about 5 years from 52 in 2000 (Interquartile range 36–67) to 57 in 2019 (IQR 40–71).

Prevalence and incidence rates were stratified by age group (0–14, 15–44, 45–64, 65+) and are shown in Fig 3 (accompanying data in S11–S16 Tables). While lifetime prevalence increased across all age groups (Fig 3A and 3B), the largest increases were seen at age 65+ (a

**Table 2. Prevalence of recorded neuromuscular disease in the UK on 1/1/2019.**

| Classification | n | All (95% CI) | Females (95%CI) | Males (95%CI) |
|---|---|---|---|---|
| **Motor Neuron Disorders** | | | | |
| • Motor neurone disease (MND) | 1,586 | 12.6 (11.9–13.2) | 10.1 (9.3–10.9) | 15.1 (14.1–16.0) |
| • Post polio syndrome | 468 | 3.7 (3.4–4.0) | 3.7 (3.2–4.2) | 3.7 (3.3–4.2) |
| • Spinal muscular atrophy | 629 | 5.0 (4.6–5.4) | 4.2 (3.7–4.7) | 5.8 (5.2–6.4) |
| **Muscle Disease** | | | | |
| *Acquired myopathies* | | | | |
| • Endocrine myopathy | 59 | 0.5 (0.3–0.6) | 0.5 (0.3–0.7) | 0.4 (0.3–0.6) |
| • Infectious myopathy | 438 | 3.5 (3.1–3.8) | 2.9 (2.5–3.3) | 4.1 (3.6–4.6) |
| • Inflammatory myopathies (IIM) | 3,152 | 25.0 (24.1–25.8) | 31.3 (30.0–32.7) | 18.6 (17.5–19.6) |
| • Toxic or drug-induced myopathy | 324 | 2.6 (2.3–2.8) | 2.0 (1.7–2.4) | 3.1 (2.7–3.5) |
| *Hereditary myopathies* | | | | |
| • Congenital myopathy | 387 | 3.1 (2.8–3.4) | 3.0 (2.5–3.4) | 3.2 (2.7–3.6) |
| • Metabolic myopathies | 709 | 5.6 (5.2–6.0) | 4.9 (4.3–5.4) | 6.4 (5.7–7.0) |
| • Muscular dystrophy (MD) | 3,723 | 29.5 (28.5–30.4) | 23.3 (22.1–24.5) | 35.7 (34.2–37.1) |
| *Other Muscle Disease* | | | | |
| • Mitochondrial disease | 563 | 4.5 (4.1–4.8) | 4.8 (4.3–5.3) | 4.1 (3.6–4.6) |
| • Myotonic disorders (non-dystrophic) | 334 | 2.6 (2.4–2.9) | 2.2 (1.8–2.6) | 3.1 (2.7–3.5) |
| • Myotonic disorders (unspecified) | 407 | 3.2 (2.9–3.5) | 3.4 (3.0–3.9) | 3.0 (2.6–3.4) |
| • Periodic paralysis | 159 | 1.3 (1.1–1.5) | 0.8 (0.6–1.1) | 1.7 (1.4–2.0) |
| **Neuropathies** | | | | |
| *Hereditary neuropathy* | | | | |
| • Charcot-Marie Tooth (CMT) | 3,724 | 29.5 (28.5–30.4) | 26.8 (25.5–28.0) | 32.2 (30.8–33.6) |
| • Other | 83 | 0.7 (0.5–0.8) | 0.5 (0.4–0.7) | 0.8 (0.6–1.0) |
| *Inflammatory & autoimmune neuropathies* | | | | |
| • Guillain-Barré syndrome (GBS) | 5,064 | 40.1 (39.0–41.2) | 36.3 (34.8–37.8) | 43.9 (42.3–45.6) |
| • GBS recorded in last 5 years only | 1,325 | 10.5 (9.9–11.1) | 8.8 (8.1–9.6) | 12.2 (11.3–13.0) |
| • Other | 1,656 | 13.1 (12.5–13.7) | 9.5 (8.7–10.2) | 16.8 (15.8–17.8) |
| **Neuromuscular Junction Disorders** | | | | |
| • Eaton-Lambert syndrome | 44 | 0.3 (0.2–0.5) | 0.4 (0.2–0.5) | 0.3 (0.2–0.5) |
| • Myasthenia gravis (MG) | 4,250 | 33.7 (32.7–34.7) | 33.6 (32.2–35.1) | 33.7 (32.3–35.1) |
| • Other | 401 | 3.2 (2.9–3.5) | 2.9 (2.5–3.4) | 3.4 (3.0–3.9) |
| Muscular or neuromuscular disease unspecified | 930 | 7.4 (6.9–7.8) | 7.0 (6.3–7.7) | 7.7 (7.0–8.4) |
| **All Neuromuscular Disease** | | | | |
| • Any | 28,230 | 223.6 (221.0–226.2) | 208.3 (204.8–211.9) | 239.0 (235.1–242.8) |
| • Any, standardised to 2019 mid-year UK population estimates | | 220.3 (217.7–222.9) | 206.2 (202.7–209.7) | 234.8 (231.1–238.6) |
| **All Neuromuscular Disease, only with GBS in last 5 years** | | | | |
| • Any | 24,565 | 194.6 (192.1–197.0) | 181.4 (178.1–184.7) | 207.8 (204.2–211.4) |
| • Any, standardised to 2019 mid-year UK population estimates | | 191.9 (189.5–194.3) | 179.6 (176.3–182.9) | 204.5 (201.0–208.0) |

Note: Rates are per 100,000 persons. All neuromuscular disease is additionally age-sex standardised to ONS 2019 mid-year population estimates for the whole of the UK. All rates represent lifetime prevalence, except for GBS which is additionally presented including only codes recorded in last 5 years. Patients can belong to multiple categories except for "unspecified", where they are only classified if no other appropriate classification was available. Denominators used were: 12,625,287 (All), 6,328,836 (Females) and 6,296,451 (Males).

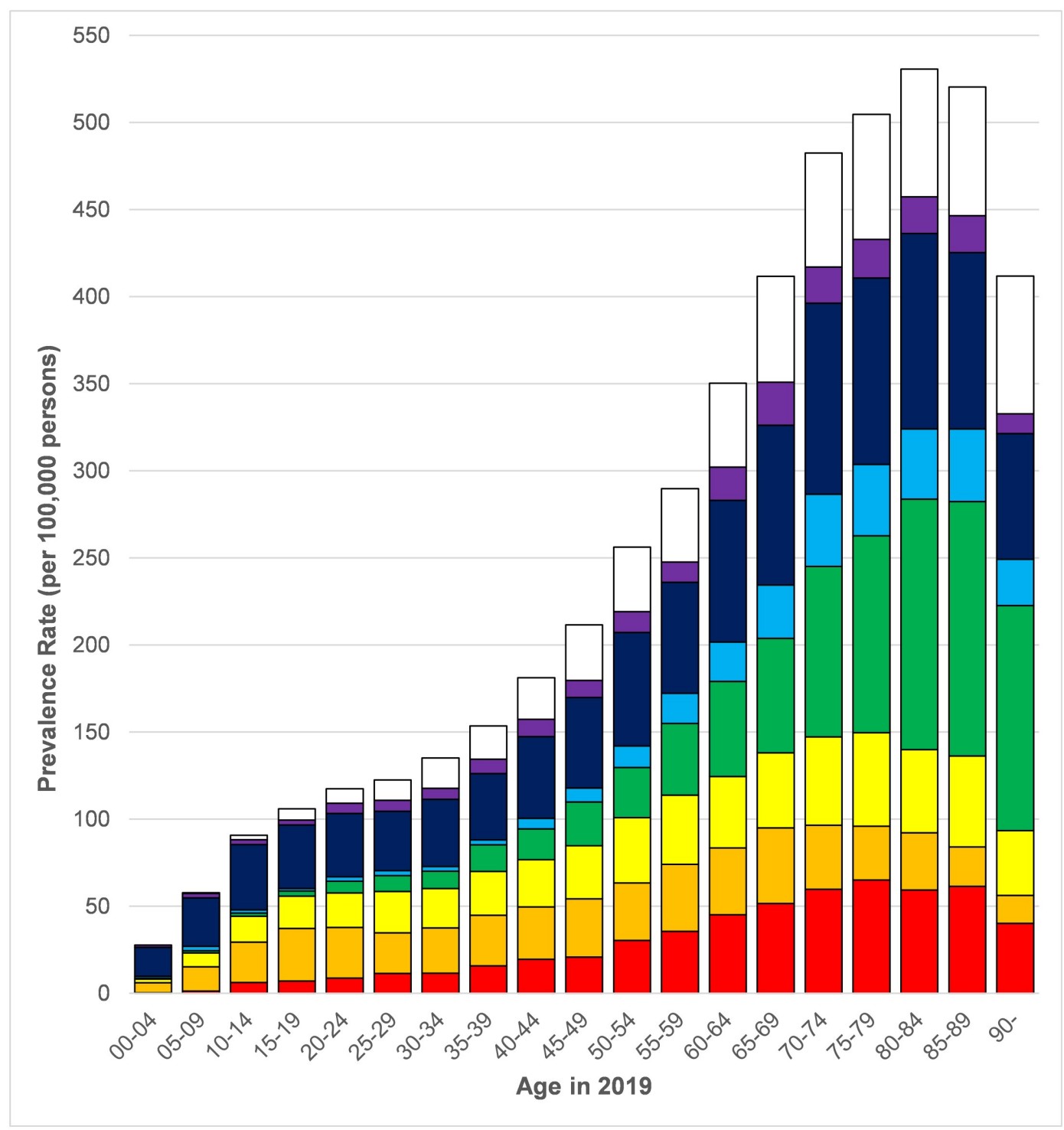

**Fig 1. Prevalence rates per 100,000 persons in 2019 by 5 year-age group and type of neuromuscular disease.** Red = Inflammatory myopathies, Orange = Muscular dystrophies, Yellow = Charcot-Marie Tooth disease, Green = Myasthenia gravis, Light Blue = Motor neurone disease, Dark Blue = Other neuromuscular disease, Purple = Guillain-Barré syndrome (recorded in last 5 years), White = Guillain-Barré syndrome (recorded last more than 5 years ago).

**Table 3. Incidence of recorded neuromuscular disease in the UK between 2015–2019.**

| Classification | Total new cases | All (95% CI) | Females (95%CI) | Males (95%CI) |
|---|---|---|---|---|
| **Motor Neuron Disorders** | | | | |
| • Motor neurone disease (MND) | 2,091 | 3.43 (3.29–3.58) | 2.91 (2.71–3.10) | 3.96 (3.74–4.19) |
| • Post polio syndrome | 77 | 0.13 (0.10–0.15) | 0.12 (0.08–0.16) | 0.13 (0.09–0.17) |
| • Spinal muscular atrophy | 118 | 0.19 (0.16–0.23) | 0.17 (0.12–0.22) | 0.22 (0.16–0.27) |
| **Muscle Disease** | | | | |
| *Acquired myopathies* | | | | |
| • Endocrine myopathy | 8 | 0.01 (0.00–0.02) | 0.02 (0.00–0.03) | 0.01 (0.00–0.02) |
| • Infectious myopathy | 127 | 0.21 (0.17–0.24) | 0.17 (0.12–0.22) | 0.25 (0.19–0.30) |
| • Inflammatory myopathies (IIM) | 790 | 1.30 (1.21–1.39) | 1.61 (1.47–1.75) | 0.98 (0.87–1.10) |
| • Toxic or drug-induced myopathy | 127 | 0.21 (0.17–0.24) | 0.15 (0.11–0.20) | 0.26 (0.21–0.32) |
| *Hereditary myopathies* | | | | |
| • Congenital myopathy | 127 | 0.21 (0.17–0.24) | 0.23 (0.17–0.28) | 0.19 (0.14–0.24) |
| • Metabolic myopathies | 147 | 0.24 (0.20–0.28) | 0.19 (0.14–0.24) | 0.30 (0.24–0.36) |
| • Muscular dystrophy (MD) | 705 | 1.16 (1.07–1.24) | 0.89 (0.79–1.00) | 1.43 (1.29–1.56) |
| *Other* | | | | |
| • Mitochondrial disease | 178 | 0.29 (0.25–0.34) | 0.34 (0.28–0.41) | 0.24 (0.19–0.30) |
| • Myotonic disorders (non-dystrophic) | 73 | 0.12 (0.09–0.15) | 0.16 (0.11–0.20) | 0.08 (0.05–0.11) |
| • Myotonic disorders (unspecified) | 132 | 0.22 (0.18–0.25) | 0.23 (0.18–0.28) | 0.20 (0.15–0.25) |
| • Periodic paralysis | 23 | 0.04 (0.02–0.05) | 0.02 (0.00–0.03) | 0.06 (0.03–0.09) |
| **Neuropathies** | | | | |
| *Hereditary neuropathy* | | | | |
| • Charcot-Marie Tooth (CMT) | 879 | 1.44 (1.35–1.54) | 1.25 (1.13–1.38) | 1.64 (1.49–1.78) |
| • Other hereditary neuropathy | 13 | 0.02 (0.01–0.03) | 0.01 (0.00–0.03) | 0.03 (0.01–0.05) |
| *Inflammatory & autoimmune neuropathies* | | | | |
| • Guillain-Barré syndrome (GBS) | 1,040 | 1.71 (1.60–1.81) | 1.39 (1.26–1.52) | 2.03 (1.87–2.19) |
| • Other inflammatory & autoimmune neuropathies | 429 | 0.70 (0.64–0.77) | 0.45 (0.38–0.53) | 0.96 (0.85–1.07) |
| **Neuromuscular Junction Disorders** | | | | |
| • Eaton-Lambert syndrome | 21 | 0.03 (0.02–0.05) | 0.04 (0.02–0.06) | 0.03 (0.01–0.05) |
| • Myasthenia gravis (MG) | 1,501 | 2.46 (2.34–2.59) | 2.06 (1.90–2.22) | 2.87 (2.68–3.06) |
| • Other NMJ disorder‡ | 76 | 0.12 (0.10–0.15) | 0.11 (0.08–0.15) | 0.13 (0.09–0.18) |
| Muscular or neuromuscular disease unspecified* | 312 | 0.51 (0.46–0.57) | 0.47 (0.40–0.55) | 0.55 (0.47–0.63) |
| **All Neuromuscular Disease** | | | | |
| • Any | 8,563 | 14.09 (13.79–14.38) | 12.35 (11.96–12.75) | 15.83 (15.38–16.28) |
| • Any, standardised to 2019 mid-year UK population estimates | | 14.22 (13.92–14.53) | 12.47 (12.07–12.87) | 16.02 (15.57–16.47) |

Note: Rates are the estimated rate calculated from 2015–9 data per 100,000 person years. All neuromuscular disease is additionally age-sex standardised to ONS 2019 mid-year population estimates for the whole of the UK. Patients can belong to multiple categories except for "unspecified", where they are only classified if no other appropriate classification was available. Denominators used were (Any of the Above): 60,790,383 (All), 30,475,599 (Females), 30,314,785 (Males).

98% increase for females over the study period, 94% for men). The pattern for incidence was different (Fig 3C and 3D), with youngest age groups (0–14, 15–44) generally unchanged over time. In the 45–64 group, there was some suggestion that it may have fallen slightly over time (by about 15% for women between 2000 and 2019, 12% for men). In the oldest age group (65 +), both sexes looked to be on an upward trajectory, but more so among women (38% increase vs. 10% for men between 2000 and 2019).

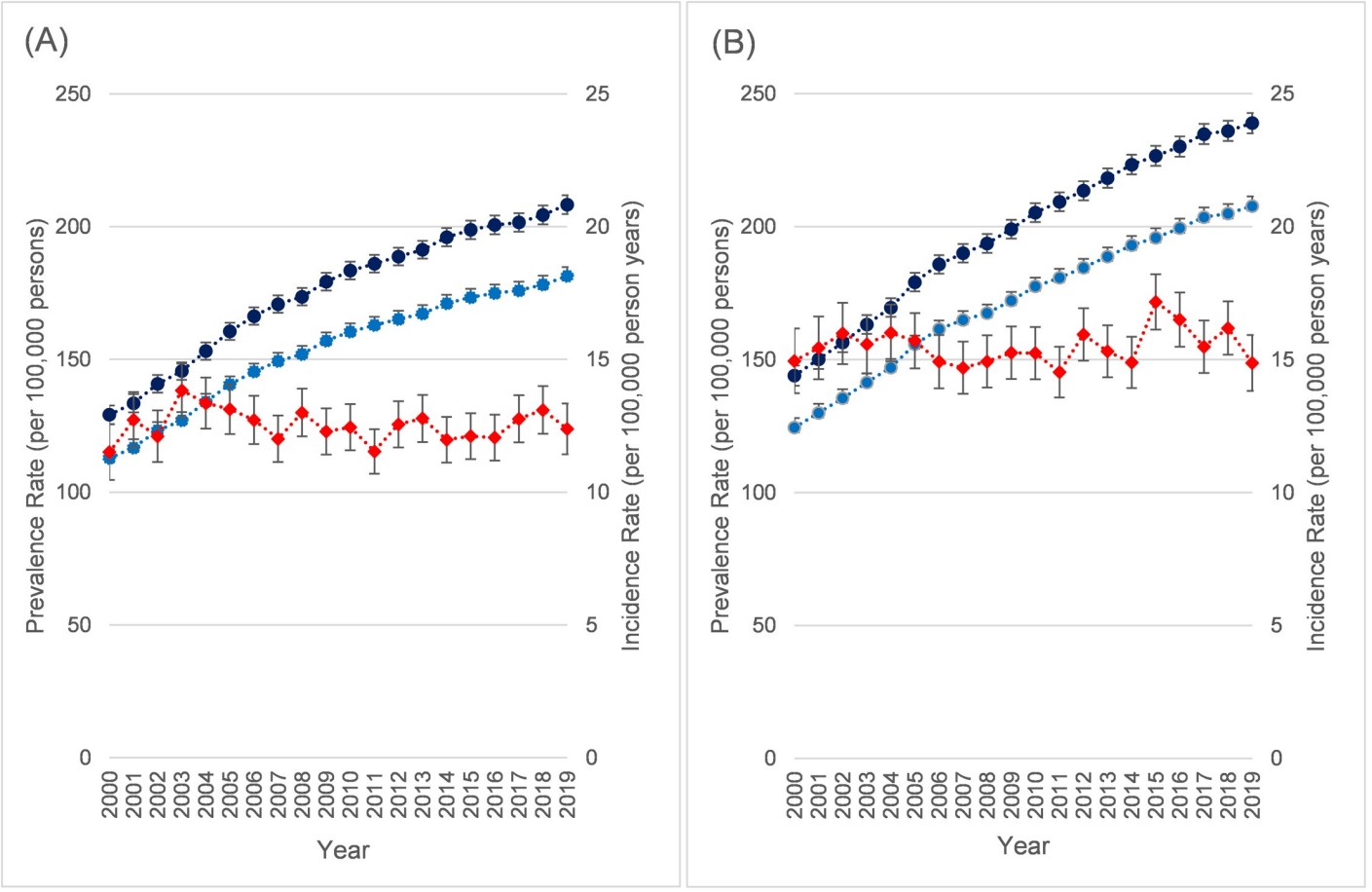

**Fig 2. Prevalence and incidence rates per 100,000 persons in neuromuscular disease 2000–2019.** (A) Females. (B) Males. Dark Blue = Lifetime prevalence rate per 100,000 persons. Light Blue = Lifetime prevalence rate per 100,000 persons excluding Guillain-Barré syndrome codes not recorded in previous 5 years. Red = Incidence rate per 100,000 person-years. Both rates age standardised to CPRD population as of 1/1/2019. 95% confidence intervals shown for each.

## Time trends in selected conditions

Annual trends in prevalence and incidence by the 6 most common conditions for males and females combined are shown in Fig 4 (accompanying data in S17 and S18 Tables). For lifetime prevalence, steady increases are seen for all conditions except MD (Fig 4B) and MND (Fig 4F) which are both largely unchanged since about 2010. For GBS, prevalence was still rising when restricted to patients with a code in the previous 5 years. For incidence, the clearest evidence of a steady increase is seen with MG (Fig 4E) which has risen since 2008, while for MD (Fig 4B) overall incidence was initially higher, but appears to reduce from about 2012 onwards. The prevalence rates were higher by 2019 than 2000 for all conditions, but the nature of the increase was not uniform. CMT disease (Fig 4C) showed the greatest rise from 12.3 per 100,000 in 2000 to 29.2 per 100,000, while MND (Fig 4F) was only marginally greater over the same period (12.6 vs. 11.1). The prevalence of MD (Fig 4B) appeared to plateau around 2006 and was steady at around 29–30 per 100,000 from that point onwards.

The annual trends in prevalence and incidence for the 6 selected conditions were summarised among 0–44-year-olds (Fig 5) and 45+ year olds (Fig 6), with different scales in each figure for clarity (accompanying data in S19–S22 Tables). Among younger age groups, the rise in

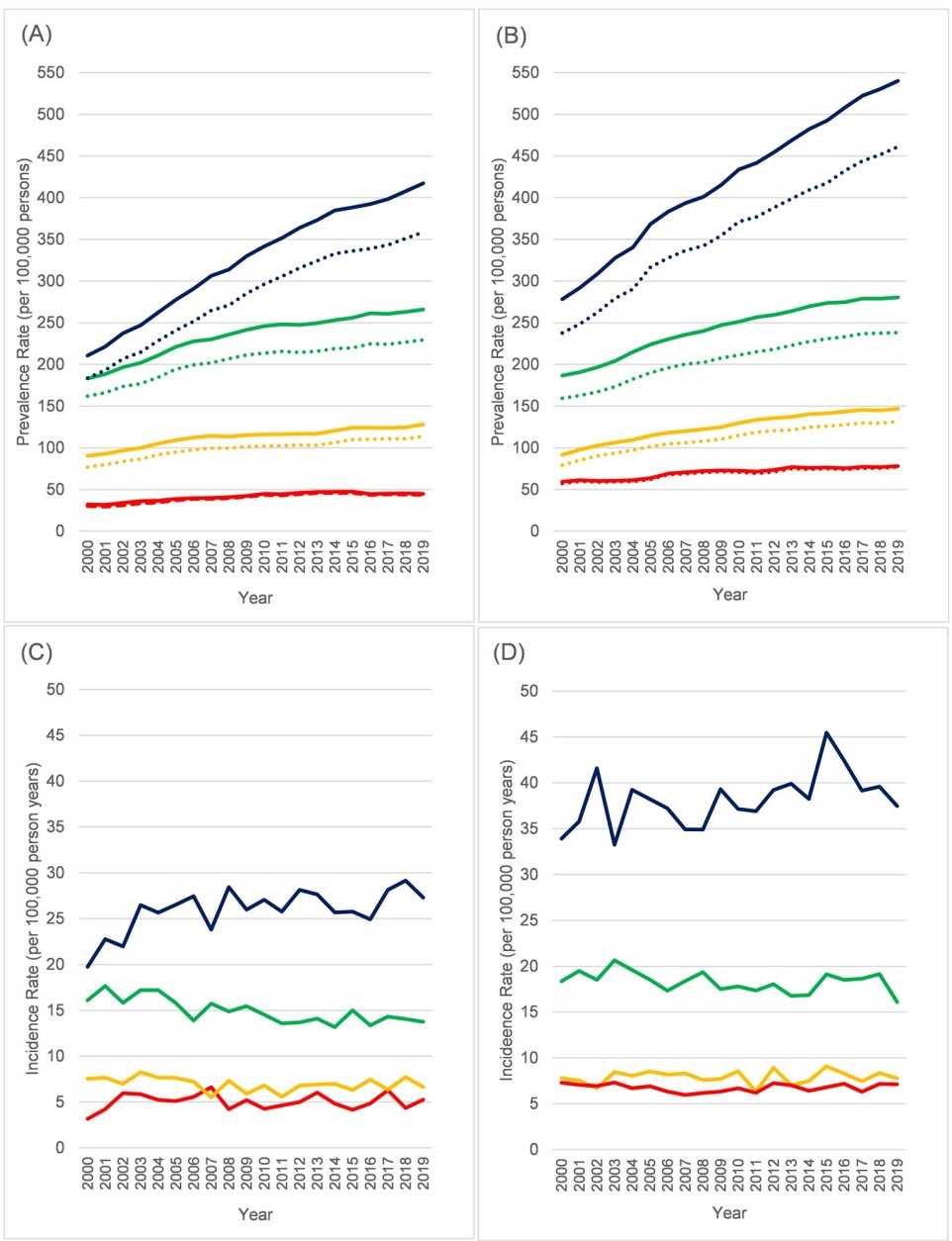

**Fig 3. Prevalence and incidence rates (per 100,000 persons) in neuromuscular disease 2000–2019 by age.** (A) Prevalence in females. (B) Prevalence in males. (C) Incidence in females. (D) Incidence in males. Red = 0–14 years. Orange = 15–44 years. Green = 45–64 years. Blue = 65+ years. For prevalence, Solid lines = Lifetime, Dotted lines = excluding Guillain-Barré syndrome codes not recorded in last 5 years. All rates age standardised (within age-group) to CPRD population as of 1/1/2019.

CMT prevalence is clearly seen (Fig 5A and 5B). In 0–14-year-olds, the prevalence of recorded MD has fallen over the study period (Fig 5A). Trends in incidence are hard to discern due to small numbers (Fig 5C–5D), though CMT disease in 0–14-year-olds was becoming more common relative to the other conditions except MD (Fig 5C). Among older age groups, prevalence of CMT and MG steadily grew among 45-64-year-olds (Fig 6A), while incidence was generally

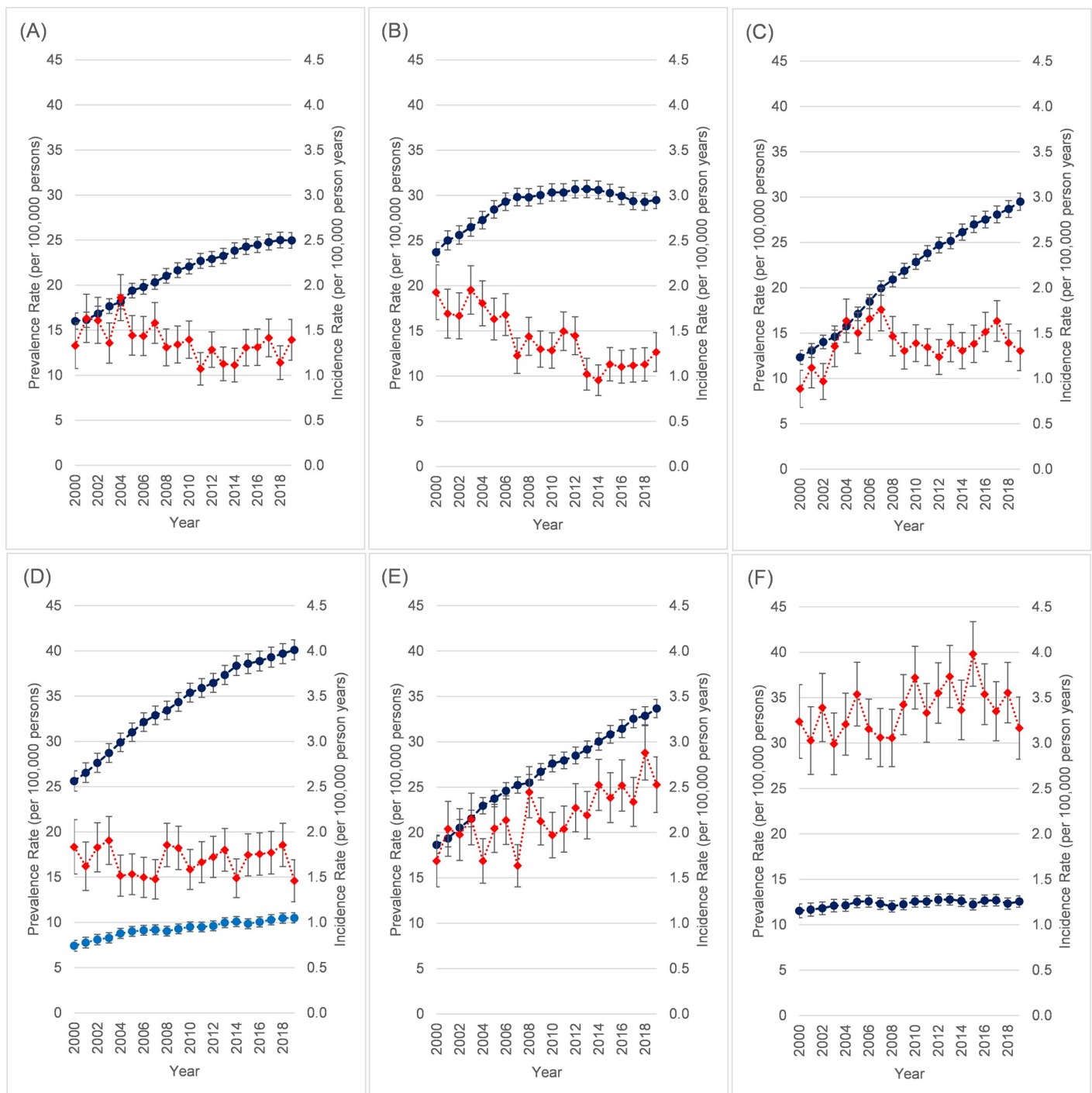

**Fig 4. Time trends in neuromuscular disease 2000–2019 by selected condition.** (A) Inflammatory myopathy. (B) Muscular dystrophy. (C) Charcot-Marie Tooth. (D) Guillain-Barre syndrome. (E) Myasthenia gravis. (F) Motor neurone disease. Dark Blue = Lifetime prevalence rate per 100,000 persons. Light Blue = Lifetime prevalence rate per 100,000 persons excluding Guillain-Barré syndrome codes not recorded in previous 5 years (D only). Red = Incidence rate per 100,000 person-years. Both rates age standardised to CPRD population as of 1/1/2019. 95% confidence intervals shown for each.

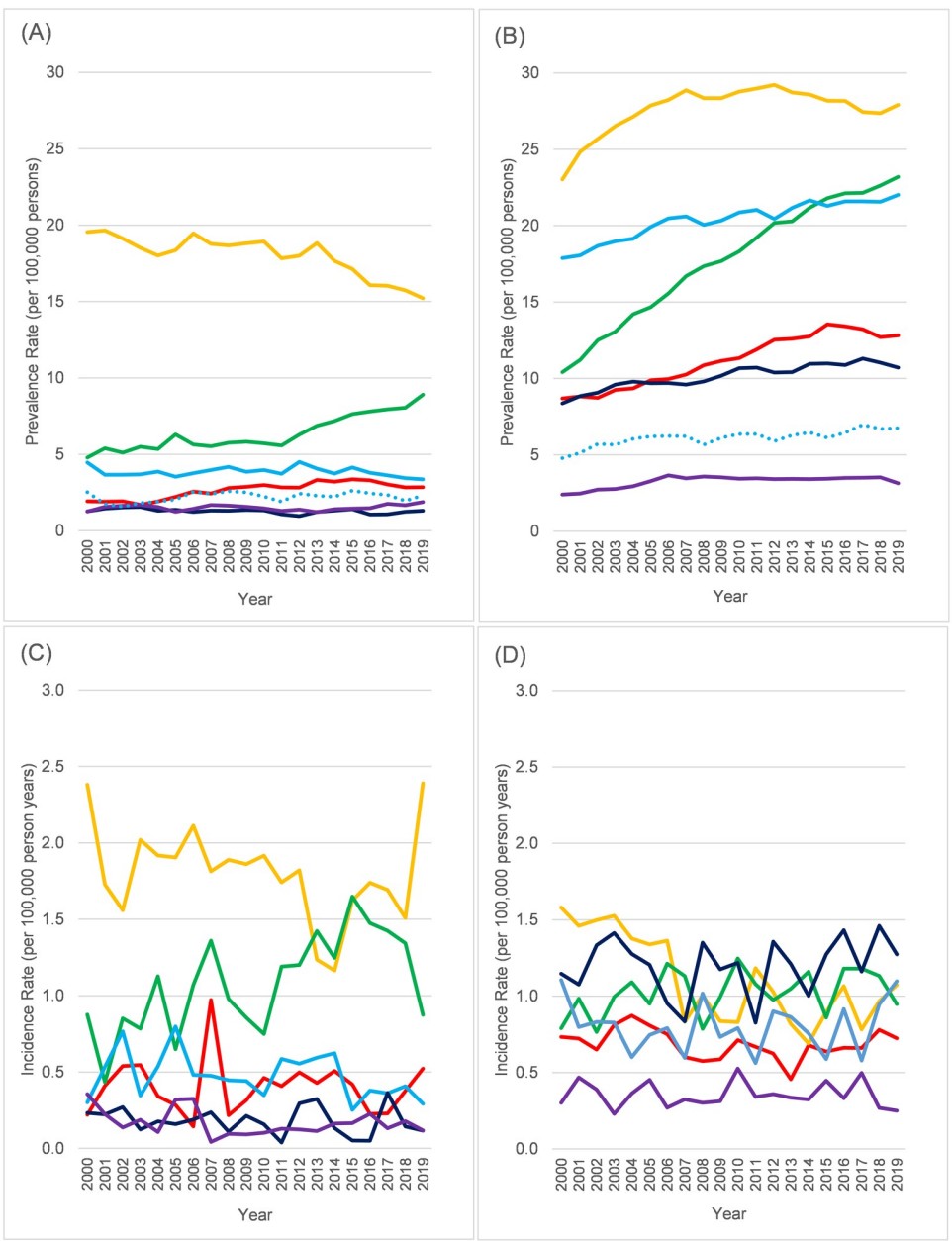

**Fig 5. Time trends in neuromuscular disease 2000–2019 by selected condition, ages 0–44 only.** (A) Prevalence in ages 0–14 years. (B) Prevalence in ages 15–44. (C) Incidence in ages 0–14 years. (D) Incidence in ages 15–44 years. Red = Inflammatory myopathies, Orange = Muscular dystrophies, Green = Charcot-Marie Tooth disease, Light Blue = Guillain-Barré syndrome, Dark Blue = Myasthenia gravis, Purple = Motor neurone disease. Solid lines = Lifetime, Dotted lines = excluding Guillain-Barré syndrome codes not recorded in last 5 years. All rates age standardised (within age-group) to CPRD population as of 1/1/2019.

unchanged over time for all conditions in this age group (Fig 6C). Among 65+ year olds, there were dramatic increases between 2000 and 2019 in ever having been diagnosed with MG (45.9 to 105.3 per 100,000) and GBS (47.0 to 91.6 per 100,000), though the largest relative rise in prevalence in this age-group was seen in CMT disease which more than tripled from 16.0 to 49.0 per 100,000. This was accompanied by more than a doubling in CMT incidence (Fig 6D)

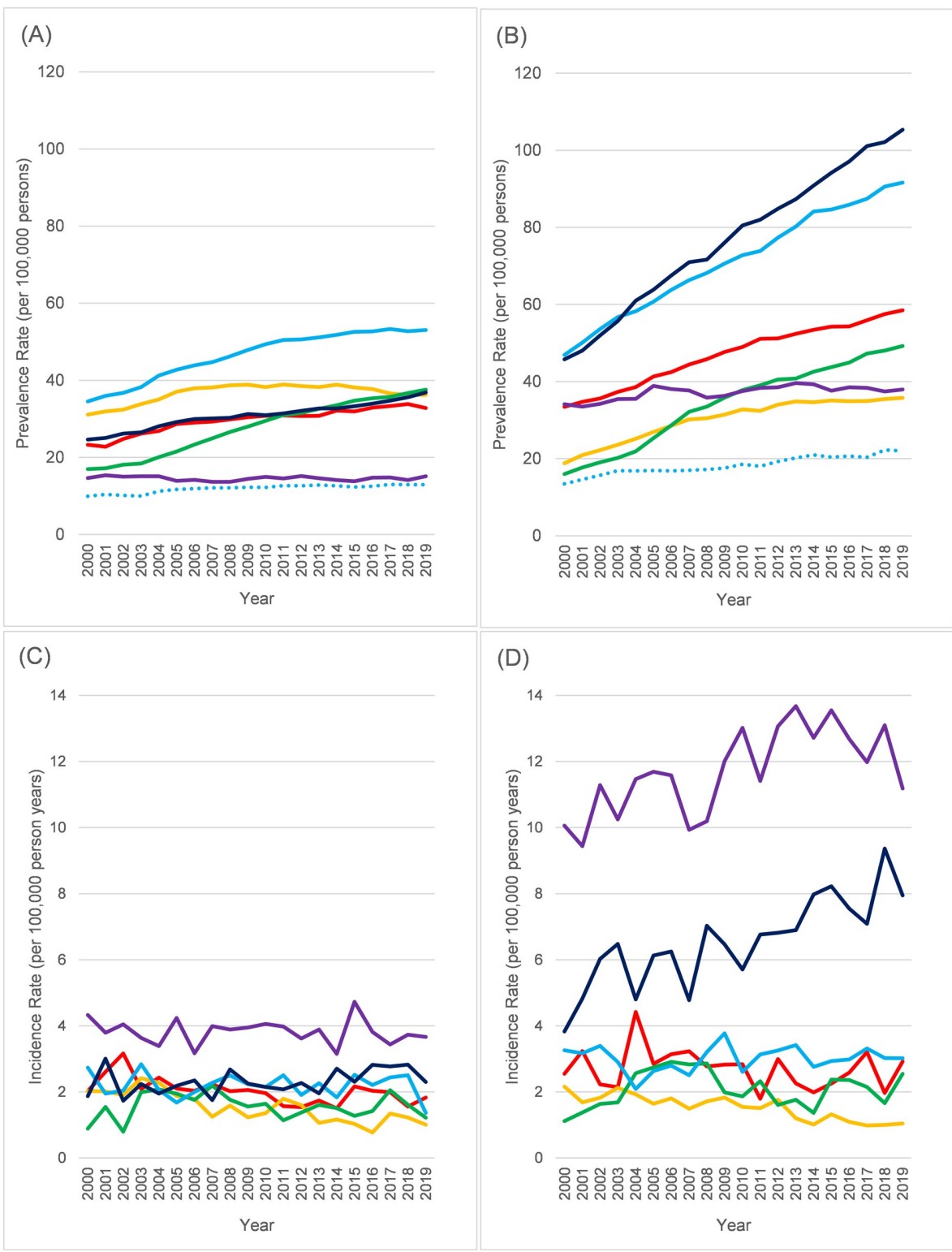

**Fig 6. Time trends in neuromuscular disease 2000–2019 by selected condition, ages 45–99 only.** (A) Prevalence in ages 45–64 years. (B) Prevalence in ages 65–99. (C) Incidence in ages 45–64 years. (D) Incidence in ages 65–99 years. Red = Inflammatory myopathies, Orange = Muscular dystrophies, Green = Charcot-Marie Tooth disease, Light Blue = Guillain-Barré syndrome, Dark Blue = Myasthenia gravis, Purple = Motor neurone disease. Solid lines = Lifetime, Dotted lines = excluding Guillain-Barré syndrome codes not recorded in last 5 years. All rates age standardised (within age-group) to CPRD population as of 1/1/2019.

over the same period (3.8 to 7.9 per 100,000). Only MND did not show a large rise in prevalence among 65+ year olds, though incidence rates were higher post-2010.

## Discussion

Using a large representative electronic primary care database, we have estimated trends over the last 20 years in the recorded prevalence and incidence of NMD in the UK. As existing estimates of NMD globally are usually based on secondary or tertiary healthcare settings without direct denominator data, our novel approach to focus on recordings from primary care gives a new perspective on the burden of NMD in the wider population. While we observed generally stable trends over time in prevalence for conditions with reduced life expectancy (MD, MND), the prevalence for other neuromuscular conditions, such as CMT and MG, has steadily grown over time.

### Overall NMD estimates

In the UK, a landmark 2010 report by the Muscular Dystrophy Campaign (MDC) estimated that approximately 71,000 patients were living with a NMD [17], representing a prevalence of about 120 per 100,000 for the UK population at the time. While the report relied on the earlier population study of 1,100 patients with genetic muscle disease in northern England [8], it also used other historical and international historical data sources in its estimation. Our analysis produced a higher estimate of lifetime prevalence throughout, rising from 136 per 100,000 in 2000 to 224 per 100,000 by 2019. Standardising to mid-year population estimates for 2019, this suggests approximately 147,000 people in the UK may have ever received a diagnosis for a NMD in their lifetime, with approximately 9,500 newly diagnosed in a single year (S23 Table). Excluding patients with no recent recording of GBS lowers the estimate to 128,000, representing a similar number for other neurological diseases such as multiple sclerosis [18] and Parkinson's disease [19], based on other recent studies of UK primary care data (S24 Table).

   Our total prevalence figure also exceeds the 160 per 100,000 estimated by Dennen et al in their comprehensive review [5], suggesting our data may be capturing patients not recorded by other datasets or methodologies, a more inclusive definition (we included MND unlike the MDC report [17]) but may also represent real increases over these historical estimates. However, overall comparisons are problematic due to the selection of conditions that have been included or excluded from the definition of NMD. For example, recent large population studies from Ireland [20], Canada [9], Norway [21] all estimated NMD prevalence but did not count polymyositis or dermatomyositis. Additionally, the Canadian study included cerebral palsy, spina bifida and multiple sclerosis in its NMD definition so any overall comparison is meaningless [9]. The Norwegian study estimated a prevalence of 112 per 100,000 for hereditary neuromuscular disorders in 2020 [21], more similar to our estimate if we excluded acquired myopathies and myasthenia gravis. The Irish study estimated a lower overall adult prevalence of 63 per 100,000 in 2013, possibly due to a stricter case definition that used multiple data sources and excluded patients labelled with possible myopathies or neuropathies [20]. Finally, a recent study from Hong Kong of all rare neurological diseases estimated a prevalence of 38 per 100,000 for NMD, but GBS and MG were classified as neuroimmune rather than neuromuscular rare neurological diseases [22]. Overall, these inconsistencies led us to focus more on comparing published estimates for the more common NMD conditions.

### Individual NMD estimates

**Motor neurone disease.** Motor neurone disease (MND) describes a group of rare neurodegenerative disorders affecting the motor neurones in the brain and spinal cord, of which

Amyotrophic Lateral Sclerosis (ALS) is the most common and well-known [23]. There have been several large systematic reviews and meta-analyses for ALS [24–26], while the broader group of MND was included in *The Lancet* Global Burden of Disease (GBD) Study 2016, which analysed cause of death data to model estimate [23]. Higher prevalence estimates in Europe have been reported than elsewhere (9.6 for ALS [26], 12.6 for MND [23] per 100,000), and a recent study from the Netherlands using linked national datasets estimate the prevalence of MND at 9.5 per 100,000 in 2017 [27]. By comparison, our estimate based on any diagnostic code for MND by 2019 was 12.6 per 100,000. The higher prevalence in males is well established, but our 50% higher estimate was more than the 25% reported in the GBD 2016 [23]. For MND incidence, our estimate of 3.4 per 100,000 person-years during 2015–9 was higher than the GBD 2016 estimate for Western Europe (2.0), and what has been reported in recent Dutch [27] (2.6 per 100,000 between 1998–2017) and Spanish [28] studies (1.7 to 2.2 per 100,000 between 2011–19). We found little evidence of temporal trends with both incidence and prevalence generally constant over the 20-year period from 2000–19, contrasting with GBD 2016 which reported a 31.2% rise in prevalence in Western Europe when compared with 1990 estimates [23].

**Inflammatory myopathies.** Inflammatory myopathies are a group of autoimmune diseases, whose cause is unknown (commonly abbreviated as IIM to denote their idiopathic nature), but are all characterised by muscle inflammation and damage, causing progressive weakness [17]. The main classifications historically have been polymyositis (PM) and dermatomyositis (DM), although classification and terminology have evolved over time. Our reported prevalence of IIM of 25.0 (per 100,000) in 2019, was similar to the 27.2 per 100,000 estimated in England in 2009 using CPRD data previously in combination with linked hospitalisation data [29]. A systematic review in 2015 found a wide range of IIM prevalence (2.4 to 33.8 per 100,000 from 10 studies), with heterogeneity from methodological differences in their meta-analysis estimates (14 per 100,000 all, 30 per 100,000 adults) [30]. All studies consistently found higher rates in females as we did (31.3 females vs. 18.6 males) and increasing prevalence with age (58.5 for ages 65+). A limitation of our approach is that we did not distinguish whether disease is active or in remission, and we only required a diagnosis to be present for a case to be initially counted. A recent study from Sweden demonstrated how their prevalence estimate varied (from 12 to 17) depending on number of specialist visits and co-prescribing of immunosuppressants or glucocorticoids [31]. For IIM incidence, studies have historically reported rates between 0.1 and 1 per 100,000 person-years, but estimates were generally higher from studies solely based on health insurance administrative databases to define cases [30]. We reported an annual incidence rate of 1.3 during 2015–9, which compares with the previous estimate from England of 1.9 from 2000–9 [29], and a more recent estimate of 1.1 during 2007–11 using national data from Sweden [31]. Though we found no evidence of an increase in incidence over time in our study, there was a steady rise in prevalence over the study period from 16 per 100,000 in 2000 to 25 per 100,000 in 2019. This observation mirrors other large database studies that have looked at time trends in Japan [32] and Korea [33].

**Muscular dystrophy.** The term muscular dystrophy (MD) refers to a group of over 30 different genetic muscle diseases that result in progressive muscle weakness and wasting due to the degeneration of muscle cells [17]. Systematic reviews and meta-analyses have attempted to provide detailed estimates of the different and combined MDs across the literature [34–37], despite high levels of heterogeneity between estimates reflecting variability in methodologies and the classification of disease codes among the studies. In the UK, the most prominent estimates have come from studies of genetic muscle disease in Northern England [8] and Northern Ireland [38]. When these studies' estimates were restricted to the muscular dystrophies, the prevalence (per 100,000) has been estimated to be 25.5 (in 2007) and 19.2 (in 1994) in their

populations respectively [35], which compares to our estimate of 29.5 for 2019. While our data source is large and representative, a limitation of our approach is the lack of diagnostic validation. The lower incidence rates from about 2012 we observed may be a coding artefact, where there was less reliance on using non-specific Read codes in favour of codes denoting specific MD type. This change may be related to increased availability of genetic testing and improved recording. However even by 2019, about one-quarter of MD patients still only had a non-specific Read code for muscular dystrophy on their primary care record, some of which may reflect uncertainty of diagnosis or a test in progress [8]. When we investigated specific types of muscular dystrophy for recent estimates of prevalence (S3 Table), our findings were similar to what has been reported elsewhere, with myotonic dystrophy (type 1) to be the most common type of MD with a pooled prevalence estimate of 8.3 [35], similar to the 9.3 we estimated in 2019. The same review produced pooled estimates for Duchenne MD (3.5) and Becker MD (2.2) across the whole population [35], comparable to our 2019 estimates (3.0 and 2.1 respectively).

**Charcot-Marie tooth.**  Charcot-Marie-Tooth disease (CMT) is a collective term for a group of hereditary motor and sensory neuropathies that affect the peripheral nervous system. and are characterised by slowly progressive distal muscle weakness below the knees and in the hands [17]. Over our study period (2000–19) we observed a 48% increase in incidence, accompanied by a 139% increase in the prevalence, which had risen from 12.3 to 29.5 per 100,000 by 2019. Compared to other neuromuscular conditions, large-scale epidemiological data on CMT is scarce. Previous estimates of the prevalence from a systematic review of 10 studies in 2016 reported most countries with a prevalence in the range of 10–20 per 100,000 [39], so our estimates are plausible if there has been an increase in CMT over time. A recent study in Norway estimated a prevalence of 29.9 per 100,000 in 2020 [21] very similar to our latest estimate. The only evidence for a temporal trend we are aware of comes from a study in Northern England, that identified CMT patients in 2010 using multiple health databases, comparing their findings with an earlier epidemiological study from the same area [40]. They found that CMT prevalence in the region had doubled to 9.8 per 100,000, though this may be an underestimate, as it was higher in Newcastle (15.2) where case ascertainment was more complete. The doubling in prevalence was seen at all ages, so cannot simply be attributed to an ageing population. Instead, the rising incidence may be related to an increase in identification, as CMT research is a developing field, with many recent advances in genetic diagnoses and new genes being identified as causal for CMT [41]. As CMT is associated with a substantial economic burden [42], it will be important to determine whether these increases we observed in the population are real.

**Guillain-Barré syndrome.**  Guillain-Barré syndrome (GBS) is an acute inflammatory neuropathy which occurs when the immune system attacks the peripheral nerves, disrupting the signal to the muscles and causing a rapidly progressive weakness in the limbs [17]. Though some cases can be severe requiring artificial ventilation and feeding, many individuals will often make a full recovery [3], explaining why systematic reviews of GBS have exclusively focussed on incidence calculations [43, 44], as well as concerns around a causal link with influenza vaccines [45]. We estimated an incidence rate of GBS of 1.7 per 100,000 person-years during 2015–9, largely unchanged since 2000. While this would put our estimate toward the higher end of the range (0.84 to 1.91) from studies from Western Europe in a 2009 systematic review [43], more recent national studies from Denmark [46], France [47] and Finland [48] have produced similar estimates over comparable periods. A limitation of the Read codes here was that since no specific codes exists for either chronic inflammatory demyelinating polyneuropathy (CIDP) or multifocal motor neuropathy (MMN), it is possible codes for GBS are being entered for these conditions. Additionally, our reliance on a single Read code for GBS to count cases may have led to an overestimate given that a single occurrence of a GBS diagnosis

has been shown to have a low predictive value in a US health claims databases [49]. A recent French study using national hospitalisation data estimated a crude incidence rate of 2.42 per 100,000 based on a primary discharge diagnosis but showed how more restrictive case definitions can half this estimate [50]. However, a recent study using national data covering 30 years in Denmark estimated a fairly stable incidence rate of 1.77 per 100,000 person years [51], similar to what we found. The Danish study found that the incidence was 44% higher in men, again mirroring our findings (2.0 vs. 1.4), and across the previous literature [44]. Finally, recent data from the Global Burden of Diseases Study in 2019 [52] found that the age-standardised prevalence of GBS had a small positive association with the level of socio-economic development, which mirrors the trends of higher incidence and lifetime prevalence we saw with lower levels of deprivation.

**Myasthenia gravis.** Myasthenia gravis (MG) is a disorder of the neuromuscular junction, resulting in fatigable muscle weakness [17]. Although it is an antibody-mediated autoimmune disease, higher-than-expected familial rates of MG have been observed [53]. The prevalence of MG in the UK using the CPRD has been reported previously as between 30.5 and 40.1 per 100,000 for 2013 depending on therapy and treatment rules [54], which compares with our estimate of 34 per 100,000 in 2019 based on a diagnosis only. These estimates are higher than what was reported in two different systematic reviews published in 2010 [55, 56], but these relied on many older studies with high levels of heterogeneity between them. More recent studies in Europe [57, 58] and Canada [59], using national databases or registers, have provided more comparable estimates, such as a prevalence of 36.1 per 100,000 in Sweden in 2016 [57]. In terms of incidence, one of systematic reviews estimated around 3 per 100,000 person years [56], which compares favourably to our estimate of 2.5, as well as recent estimates from 2.9 in Sweden [57] of 2.4 in Poland [58]. Both reviews suggested incidence may be increasing over time, due to greater awareness and improved methods of diagnosis [56], and this would contribute to an eventual increase in prevalence [55]. More recent large database studies from Europe [57, 60–62] with longer follow-up have confirmed the increases in MG incidence and prevalence, especially among older ages, in particularly older men [62]. In our study, the prevalence of MG more than doubled in individuals over 65 years old from 46 in 2000 to 105 in 2009 (per 100,000) and suggests that MG is becoming a disease more associated with older patients [63].

## Benefits and limitations

The epidemiology of many rare diseases has often relied on administrative hospitalisation data to estimate prevalence often without direct denominator data [64]. The availability of large-scale primary care databases in the UK provides a new perspective on the burden and distribution of these conditions. These resources can address the scarcity of information about rare diseases, assisting patient advocacy groups to potentially help improve the lives of patients and their families [65], and provide appropriate allocation of resources. However, the main limitation of our study is a lack of validation of any of the diagnoses recorded in primary care. Our aim was to initially describe the patterns in recorded NMD based on an assumption that any Read codes being used for these rare conditions would represent diagnoses that have made been made outside of the primary care setting, generally from specialist settings. While the hierarchical system of Read codes have their limitations and are being replaced with the more flexible SNOMED codes [66] in UK primary care, they still provide a level of detail beyond ICD-10 codes for many conditions. As our estimates are generally higher than what has been reported both in the UK and elsewhere, and are stable over time in terms of incidence, the question is not whether these diagnoses are consistently found on electronic primary care

records, but rather how reliable they may be. Further work could investigate this by seeking corroborating information in terms of prescribing, referrals, and linked hospitalisation data, but this will require a different criterion for each NMD.

The pattern of rising prevalence against a background of steady incidence, suggests that it cannot simply be explained by the more frequent awareness and recording of these rare conditions during our study period. A more obvious explanation instead might be rising life expectancy, where improvements in treatment for Duchenne MD for example have significantly increased survival [67]. Overall, our cohort of prevalent cases in 2019 with NMD was approximately 5-years older on average than its comparator in 2000. Given the increase in risk in being diagnosed with myasthenia gravis at older age, the overall rise in the general population of people living to older age will have an impact on the rise in future cases and disease burden. A further explanation might be a "deficit" in historical diagnoses in earlier years which is never calibrated as those patients leave the database (through de-registration or death). These patients who "exit" the dataset were less likely to have a recording for a NMD than those who "enter" the dataset over time and replace them. Other temporal studies using the CPRD over similar time periods, but for more common conditions, have also observed rising prevalence with no accompanying rise in incidence for osteoarthritis [68], psoriasis [69] and diabetes [70]. The authors primarily attributed this phenomenon to a decreasing risk in mortality over the study period.

## Conclusions

We have shown that the recording of many NMDs on UK primary care records exceeds current estimates of people thought to be living with these rare conditions. Among neurological disorders, this would suggest the prevalence of all NMD in the population is similar to multiple sclerosis and Parkinson's disease. The rise in prevalence at older ages suggests that some of these conditions are now more common within an ageing population, and future health service and care planning would benefit from greater awareness.

## Supporting information

**S1 Table. Hierarchical structure of classifications based on available Read codes.**
(PDF)

**S2 Table. Read Codes used in classifying conditions.**
(PDF)

**S3 Table. Recent prevalence and incident rates for specific muscular dystrophies.**
(PDF)

**S4 Table. Prevalence rates for recorded neuromuscular disease in 2019 by age group.**
(PDF)

**S5 Table. Prevalence rates for recorded neuromuscular disease in 2019 by region.**
(PDF)

**S6 Table. Incidence rates for recorded neuromuscular disease in 2015–2019 by region.**
(PDF)

**S7 Table. Prevalence rates for recorded neuromuscular disease in 2019 by index of multiple deprivation (England only).**
(PDF)

**S8 Table. Incidence rates for recorded neuromuscular disease in 2015–2019 index of multiple deprivation (England only).**
(PDF)

**S9 Table. Age standardised prevalence rates for all neuromuscular disease 2000–19.**
(PDF)

**S10 Table. Age standardised incidence rates for all neuromuscular disease 2000–19.**
(PDF)

**S11 Table. Age standardised lifetime prevalence rates 2000–19 for all neuromuscular disease in females by age.**
(PDF)

**S12 Table. Age standardised lifetime prevalence rates 2000–19 for all neuromuscular disease in males by age.**
(PDF)

**S13 Table. Age standardised prevalence rates 2000–19 for all neuromuscular disease in females by age, excluding Guillain-Barré syndrome codes not recorded in previous 5 years.**
(PDF)

**S14 Table. Age standardised prevalence rates 2000–19 for all neuromuscular disease in males by age, excluding Guillain-Barré syndrome codes not recorded in previous 5 years.**
(PDF)

**S15 Table. Age standardised incidence rates 2000–19 for all neuromuscular disease in females by age.**
(PDF)

**S16 Table. Age standardised incidence rates 2000–19 for all neuromuscular disease in males by age.**
(PDF)

**S17 Table. Age standardised prevalence rates 2000–19 for selected conditions.**
(PDF)

**S18 Table. Age standardised incidence rates 2000–19 for selected conditions.**
(PDF)

**S19 Table. Age standardised prevalence rates 2000–19 for selected conditions, ages 0–44 only.**
(PDF)

**S20 Table. Age standardised prevalence rates 2000–19 for selected conditions, ages 45- only.**
(PDF)

**S21 Table. Age standardised incidence rates 2000–19 for selected conditions, ages 0–44 only.**
(PDF)

**S22 Table. Age standardised incidence rates 2000–19 for selected conditions, ages 45- only.**
(PDF)

**S23 Table. Population standardised prevalence (2019) and incidence (2015–9) of recorded neuromuscular disease in the UK with estimated number of people.**
(PDF)

**S24 Table. Estimated numbers of people in UK with recorded neuromuscular disease compared with published estimates for Parkinson's and multiple sclerosis using other CPRD data.**
(PDF)

**S1 Checklist.**
(DOC)

## Acknowledgments

IC conceptualised the study and curated the datasets. IC and EB carried out the formal analysis. NN developed the coding classifications. All authors (IC, EB, NN, TH, SD, UC, DG) contributed writing and editing of the submitted manuscript.

## Author Contributions

**Conceptualization:** Iain M. Carey, Niranjanan Nirmalananthan, Tess Harris, Stephen DeWilde, Derek G. Cook.

**Data curation:** Iain M. Carey, Emma Banchoff.

**Formal analysis:** Iain M. Carey, Emma Banchoff.

**Funding acquisition:** Iain M. Carey.

**Methodology:** Iain M. Carey, Niranjanan Nirmalananthan.

**Writing – original draft:** Iain M. Carey, Tess Harris, Stephen DeWilde, Umar A. R. Chaudhry, Derek G. Cook.

**Writing – review & editing:** Iain M. Carey, Niranjanan Nirmalananthan, Tess Harris, Stephen DeWilde, Umar A. R. Chaudhry, Derek G. Cook.

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
