## [Decision Letter · Decision Letter 0]

16 Nov 2021

PONE-D-21-29273Prevalence and incidence of neuromuscular conditions in the UK between 2000 and 2019: a retrospective study using primary care dataPLOS ONE

Dear Dr. Carey,

Thank you for submitting your manuscript to PLOS ONE. After careful consideration, we feel that it has merit but does not fully meet PLOS ONE’s publication criteria as it currently stands. Therefore, we invite you to submit a revised version of the manuscript that addresses the points raised during the review process.

We look forward to receiving your revised manuscript.

Kind regards,

Julie Dumonceaux

Academic Editor

PLOS ONE

2. Please amend the manuscript submission data (via Edit Submission) to include author “Umar AR Chaudhry”

Reviewers' comments:

Reviewer's Responses to Questions

**Comments to the Author**

1. Is the manuscript technically sound, and do the data support the conclusions?

Reviewer #1: Yes

2. Has the statistical analysis been performed appropriately and rigorously? 

Reviewer #1: Yes

3. Have the authors made all data underlying the findings in their manuscript fully available?

Reviewer #1: Yes

4. Is the manuscript presented in an intelligible fashion and written in standard English?

Reviewer #1: Yes

5. Review Comments to the Author

Reviewer #1: I have read with great interest the study by Carey et al. The Authors used the data available from a national primary care database to determine prevalence and incidence of neuromuscular diseases in UK.

There are some limitations with this approach like the lack of secure diagnoses in the cases analysed. However, the methodology was accurate, limitations recognised and conclusions appropriate.

The paper does not have major shortcomings and the Authors report with very high details the epidemiological data for the different Neuromuscular subgroups.

Analyses done were limited in reporting incidence and prevalence without any additional exploratory analysis that could have given a more complete picture of NMD diseases in UK and increase impact and novelty of the study.

Overall, data presented are of significance for the scientific community and provide an important resource for future studies in neuromuscular diseases.

There are minor comments to the Authors

1. The Authors mentioned that there might be care inequities depending on care services within a region and at page 14 reported lower prevalence rates in Northern Ireland.

-Can the Authors comment on this result in the discussion?

-Can the Authors expand this topic and explore whether practices with low number of NMD cases recorded cluster in specific region/county within England? This result would give an added value to the paper and help in guiding national healthcare strategy.

2. Line 310, pg 17 “which was accompanied by more than a doubling in incidence over the same period (3.8 to 7.9 per 100,000)”

I believe the values are the one represented in Figure 6 D (1.12 to 2.55) for CMT.

Can the Author check the values reported and confirm which one is correct?

3. Line 249, page 15, there is a repetition (females)

4. Figure quality can be improved. I would perhaps include figure titles in most figure panels (especially figure 4) to increase readability.

6. PLOS authors have the option to publish the peer review history of their article (what does this mean?). If published, this will include your full peer review and any attached files.

Reviewer #1: No

---

## [Author Response · Author response to Decision Letter 0]

2 Dec 2021

Reviewer #1: I have read with great interest the study by Carey et al. The Authors used the data available from a national primary care database to determine prevalence and incidence of neuromuscular diseases in UK.

There are some limitations with this approach like the lack of secure diagnoses in the cases analysed. However, the methodology was accurate, limitations recognised and conclusions appropriate.

The paper does not have major shortcomings and the Authors report with very high details the epidemiological data for the different Neuromuscular subgroups.

Analyses done were limited in reporting incidence and prevalence without any additional exploratory analysis that could have given a more complete picture of NMD diseases in UK and increase impact and novelty of the study.

Overall, data presented are of significance for the scientific community and provide an important resource for future studies in neuromuscular diseases.

* We thank the reviewer for their encouraging words and positive feedback. 

There are minor comments to the Authors

1. The Authors mentioned that there might be care inequities depending on care services within a region and at page 14 reported lower prevalence rates in Northern Ireland.

-Can the Authors comment on this result in the discussion?

*This is an interesting point that the reviewer has raised, based on a passing comment from a limited analysis by region that was originally provided in the supplement. On reflection, this analysis and accompanying text failed to convey its limitations in terms of assessing differences by region. For example, the practices in Northern Ireland, make up only about 2% of the dataset with only about 240,000 active patients and less than 500 potential prevalent neuromuscular cases in 2019. (Northern Ireland is estimated to be about 3% of the UK, so we are also under-represented in that region). 

We have extended these analyses now, splitting England into 3 areas (North, Midlands, South) and investigating the 6 most common neuromuscular conditions (inflammatory myopathies, muscular dystrophies, Charcot-Marie Tooth disease, Guillain-Barré syndrome, myasthenia gravis & motor neurone disease). Although the number for each condition in Northern Ireland is extremely small, these additional analyses show that the incidence and prevalence is not consistently lower for all conditions. On the basis of this, we feel it would be difficult to draw any conclusion.

Although we have not commented specifically on regional variation in the discussion, we have however added a comment regarding deprivation and Guillain-Barré syndrome (see next comment). 

-Can the Authors expand this topic and explore whether practices with low number of NMD cases recorded cluster in specific region/county within England? This result would give an added value to the paper and help in guiding national healthcare strategy.

* As indicated above we have expanded these exploratory analyses further within the supplemental material (new Tables S5 to S8). We have removed the previous text, and now added the following text to the results in its place.

“Variations by region and deprivation

We explored variation in recent prevalence and incidence rates by region (Tables S5-S6) and IMD (Tables S7-S8). There was no consistent pattern by region, such that regions with the highest or lowest overall prevalence or incidence – reported some conditions higher with others being lower than the rest of the UK. Although numbers of cases were small in some of these regions, there were no obvious outliers in terms of incidence or prevalence rates. For deprivation in England (using IMD), the most consistent pattern was with Guillain-Barré syndrome, where both recent incidence (23% higher than expected) and recorded lifetime prevalence (8% higher) was greatest in the least deprived group.”

---

And we have added the following to the discussion,

“Finally, recent data from the Global Burden of Diseases Study in 2019* found that the age-standardised prevalence of GBS had a small positive association with the level of socio-economic development, which mirrors the trends of higher incidence and prevalence we saw with lower levels of deprivation.” 

* - Bragazzi, N. L., et al. (2021). "Global, regional, and national burden of Guillain–Barré syndrome and its underlying causes from 1990 to 2019." Journal of Neuroinflammation 18(1).

-----

We think the issue of practice clustering is only relevant here when checking the codes for data errors and anomalies. For example, we already noted in lines 133-135 that we had to exclude 3 practices when considering the codes for medium-chain acyl-CoA dehydrogenase deficiency (MCADD) as their counts were unusually high. As each of the neuromuscular conditions are extremely rare, it would seem entirely feasible that many practices may not have any patients recorded with that specific condition. 

2. Line 310, pg 17 “which was accompanied by more than a doubling in incidence over the same period (3.8 to 7.9 per 100,000)”

I believe the values are the one represented in Figure 6 D (1.12 to 2.55) for CMT.

Can the Author check the values reported and confirm which one is correct?

* The reviewer is quite correct, and we thank them for their sharp eye in spotting this typo. We were wanting to reference the rise in incidence from 3.8 to 7.9 which was for MG not CMT. Upon reflection, we thought this section could be structured better and have re-written it.

“Among older age groups, prevalence of CMT and MG steadily grew among 45-64-year-olds (Fig 6A), while incidence was generally unchanged over time for all conditions in this age group (Fig 6C). Among 65+ year olds, there were dramatic increases between 2000 and 2019 in ever having been diagnosed with MG (45.9 to 105.3 per 100,000) and GBS (47.0 to 91.6 per 100,000), though the largest relative rise in prevalence in this age-group was seen in CMT disease which more than tripled from 16.0 to 49.0 per 100,000. This was accompanied by more than a doubling in CMT incidence (Fig 6D) over the same period (3.8 to 7.9 per 100,000). Only MND did not show a large rise in prevalence among 65+ year olds, though incidence rates were higher post-2010.”

3. Line 249, page 15, there is a repetition (females)

* It now reads “all NMD for females (Fig 2A) and males (Fig 2B) separately”

4. Figure quality can be improved. I would perhaps include figure titles in most figure panels (especially figure 4) to increase readability.

*The figures were put through the PACE software as per instructions, but we agree they looked sub-optimal on the generated PDF. We will re-generate at 600 dpi and see if this improves them. 

We like the suggestion of the labels but the submission instructions were “Do not include author names, article title, or figure number/title/caption within figure files”.

---

## [Editor Report · Decision Letter 1]

15 Dec 2021

Prevalence and incidence of neuromuscular conditions in the UK between 2000 and 2019: a retrospective study using primary care data

PONE-D-21-29273R1

Dear Dr. Carey,

We’re pleased to inform you that your manuscript has been judged scientifically suitable for publication and will be formally accepted for publication once it meets all outstanding technical requirements.

Kind regards,

Julie Dumonceaux

Academic Editor

PLOS ONE

---

## [Editor Report · Acceptance letter]

17 Dec 2021

PONE-D-21-29273R1 

Prevalence and incidence of neuromuscular conditions in the UK between 2000 and 2019: a retrospective study using primary care data 

Dear Dr. Carey:

I'm pleased to inform you that your manuscript has been deemed suitable for publication in PLOS ONE. Congratulations! Your manuscript is now with our production department. 

Kind regards, 

on behalf of

Dr. Julie Dumonceaux 

Academic Editor

PLOS ONE